# MULTI-TASK LEARNING WITH 3D-AWARE REGULARIZATION

**Wei-Hong Li[1], Steven McDonagh[1], Ales Leonardis[2], Hakan Bilen[1]**
[1]University of Edinburgh, [2]University of Birmingham
github.com/VICO-UoE/3DAwareMTL

## ABSTRACT

Deep neural networks have become the standard solution for designing models that can perform multiple dense computer vision tasks such as depth estimation and semantic segmentation thanks to their ability to capture complex correlations in high dimensional feature space across tasks. However, the cross-task correlations that are learned in the unstructured feature space can be extremely noisy and susceptible to overfitting, consequently hurting performance. We propose to address this problem by introducing a structured 3D-aware regularizer which interfaces multiple tasks through the projection of features extracted from an image encoder to a shared 3D feature space and decodes them into their task output space through differentiable rendering. We show that the proposed method is architecture agnostic and can be plugged into various prior multi-task backbones to improve their performance; as we evidence using standard benchmarks NYUv2 and PASCAL-Context.

## 1 INTRODUCTION

Learning models that can perform multiple tasks coherently while efficiently sharing computation across tasks is the central focus of multi-task learning (MTL) (Caruana, 1997). Deep neural networks (DNNs), which have become the standard solution for various computer vision problems, provide at least two key advantages for MTL. First, they allow for sharing a significant portion of features and computation across multiple tasks, hence they are computationally efficient for MTL. Second, thanks to their hierarchical structure and high-dimensional representations, they can capture complex cross-task correlations at several abstraction levels (or layers).

Yet designing multi-task DNNs that perform well in all tasks is extremely challenging. This often requires careful engineering of mechanisms that allow for the sharing of relevant features between tasks, while also maintaining task-specific features. Many recent multi-task DNNs (Vandenhende et al., 2021) can be decomposed into shared feature encoder across all tasks and following task-specific decoders to generate predictions. The technical challenge here is to strike a balance between the portion of the shared and task-specific features to achieve good performance-computation trade-off. To enable more flexible feature sharing and task-specific adaptation, Liu et al. (2019) propose to use 'soft' task-specific attention modules appended to the shared encoder that effectively shares most features and parameters across the tasks while adapting them to each task through light-weight attention modules. However, these attention modules are limited to share features across tasks only within each layer (or scale). Recent works (Vandenhende et al., 2020b; Bruggemann et al., 2021) propose to aggregate features from different layers and to capture cross-task relations from the multi-scale features. More recently, capturing long-range spatial correlations across multiple tasks has been shown to further improve MTL performance through use of vision transformer modules (Ye & Xu, 2022).

Our central hypothesis is that the high-dimensional and unstructured features, shared across tasks, in the recent MTL models are prone to capturing noisy cross-task correlations and hence hurt performance. To this end, we propose regulating the feature space of shared representations by introducing a structure that is valid for all considered tasks. In particular, we look at dense prediction computer vision problems such as monocular depth estimation, semantic segmentation where each input pixel is associated with a target value, and represent their shared intermediate features in a 3D-aware feature space by leveraging recent advances in 3D modeling and differentiable rendering (Niemeyer et al.,

2020; Mildenhall et al., 2020; Chan et al., 2022; 2023). *Our key intuition is that the physical 3D world affords us inherent and implicit consistency between various computer vision tasks.* Hence, by projecting high-dimensional features to a structured 3D-aware space, our method eliminates multiple geometrically-inconsistent cross-task correlations.

To this end, we propose a novel regularization method that can be plugged into diverse prior MTL architectures for dense vision problems including both convolutional (Vandenhende et al., 2020b) and transformer (Ye & Xu, 2022) networks. Prior MTL architectures are typically composed of a shared feature extractor (encoder) and multiple task-specific decoders. Our regularizer, instantiated as a deep network, connects to the output of the shared feature encoder, maps the encodings to three groups of feature maps and further uses these to construct a tri-plane representing planes $x-y$, $x-z$, $y-z$ as in Chan et al. (2022). We are able to query any 3D position by projecting it onto the tri-plane and retrieve a corresponding feature vector through bi-linear interpolation across the planes, passing them through light-weight, task-specific decoders and then rendering the outputs as predictions for each task by raycasting, as in Mildenhall et al. (2020). Once the model has been optimized by minimizing each task loss for both the base model and regularizer, the regularizer is removed. Hence our method does not bring any additional inference cost. Importantly, the regularizer does not require multiple views for each scene and learns 3D-aware representations from a single view. Additionally, our model generalizes to unseen scenes, as the feature encoder is shared across different scenes.

Our method relates to both MTL and 3D modeling work. It is orthogonal to recent MTL contributions that focus rather on designing various cross-task interfaces (Vandenhende et al., 2020a; Liu et al., 2019), or optimization strategies that may obtain more balanced performance across tasks (Kendall et al., 2018; Chen et al., 2018). Alternatively, our main focus is to learn better MTL representations by enforcing 3D structure upon them, through our 3D-aware regularizer. Our method can be incorporated to several recent MTL methods and improve their performance. Most related to ours, Zhi et al. (2021) and Kundu et al. (2022) extend the well-known neural radiance field (NeRF) (Mildenhall et al., 2020) to semantic segmentation and panoptic 3D scene reconstruction, respectively. Unlike them, our main focus is to jointly perform multiple tasks that include depth estimation, boundary detection, surface normal estimation, in addition to semantic segmentation. Uniquely, our method does not require multiple views. Finally, our method is not scene-specific, can learn multiple scenes in a single model and generalizes to unseen scenes.

To summarize, our main contribution is a novel 3D-aware regularization method for the MTL of computer vision problems. Our method is architecture agnostic, does not bring any additional computational cost for inference, and yet can significantly improve the performance of state-of-the-art MTL models as evidenced under two standard benchmarks; NYUv2 and PASCAL-Context.

## 2 RELATED WORK

**Multi-task Learning** MTL (Caruana, 1997) commonly aims to learn a single model that can accurately generate predictions for multiple desired tasks, given an input (see Fig. 1 (a)). We refer to Ruder (2017); Zhang & Yang (2017); Vandenhende et al. (2021) for comprehensive literature review. The prior works in computer vision problems can be broadly divided into two groups. The first group (Kokkinos, 2017; Ruder et al., 2019; Vandenhende et al., 2020a; Liang et al., 2018; Bragman et al., 2019; Strezoski et al., 2019; Xu et al., 2018; Zhang et al., 2019; Bruggemann et al., 2021; Bilen & Vedaldi, 2016; Zhang et al., 2018; Xu et al., 2018) focuses on improving network architecture via more effective information sharing across tasks by designing cross-task attention mechanisms Misra et al. (2016), task-specific attention modules (Liu et al., 2019; Bhattacharjee et al., 2023), cross-tasks feature interaction (Ye & Xu, 2022; Vandenhende et al., 2020b), gating strategies or mixture of experts modules (Bruggemann et al., 2020; Guo et al., 2020; Chen et al., 2023; Fan et al., 2022; Ye & Xu, 2023b), visual prompting (Ye & Xu, 2023a; Liu et al., 2023). The second group aims to address the unbalanced optimization for joint minimization of multiple task-specific loss functions, where each may exhibit varying characteristics. This is achieved through either actively changing loss term weights (Kendall et al., 2018; Liu et al., 2019; Guo et al., 2018; Chen et al., 2018; Lin et al., 2019; Sener & Koltun, 2018; Liu et al., 2021b), modifying the gradients of loss functions w.r.t. shared network weights to alleviate task conflicts (Yu et al., 2020; Liu et al., 2021a; Chen et al., 2020; Chennupati et al., 2019; Suteu & Guo, 2019) or knowledge distillation (Li & Bilen, 2020; Li

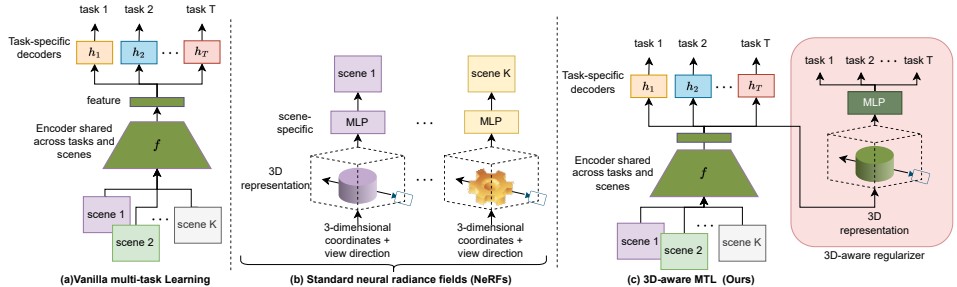

Figure 1: Illustration of (a) vanilla multi-task learning, (b) standard neural radiance fields (NeRFs) and (c) our 3D-aware multi-task learning method.

et al., 2023). Unlike these methods, our work aims to improve MTL performance by regularizing DNNs through the introduction of 3D-aware representations (see Fig. 1 (c)).

**Neural Rendering** Our approach also relates to the line of work that learns a 3D scene representations for performing novel view synthesis (Lombardi et al., 2019; Meshry et al., 2019; Sitzmann et al., 2019; Thies et al., 2019; Mildenhall et al., 2020; Chan et al., 2021; Cai et al., 2022; Gu et al., 2021), creating explicit 3D object (Lin et al., 2023), view editing (Zhang et al., 2023). Prior methods with few exceptions can represent only a single scene per model, require many calibrated views, or are not able to perform other tasks than novel view synthesis such as semantic segmentation, depth estimation (see Fig. 1 (b)). PixelNeRF (Yu et al., 2021) conditions a NeRF (Mildenhall et al., 2020) on image inputs through an encoder, allows for the modeling of multiple scenes jointly and generalizes to unseen scenes, however, the work focuses only on synthesizing novel views. Zhi et al. (2021) extend the standard NeRF pipeline through a parallel semantic segmentation branch to jointly encode semantic information of the 3D scene, and obtain 2D segmentations by rendering the scene for a given view using raycasting. However, their model is scene-specific and does not generalize to unseen scenes. Panoptic Neural Fields (Kundu et al., 2022) predict a radiance field that represents the color, density, instance and category label of any 3D point in a scene through the combination of multiple encoders for both background and each object instance. The work was designed for predicting those tasks only on novel views of previously seen scenes, hence it cannot be applied to new scenes without further training on them and is also limited to handle only rigid objects ($c.f.$ non-rigid, deformable). In contrast, our method can be used to efficiently predict multiple tasks in novel scenes, without any such restrictions on object type, can be trained from a single view and is further not limited to a fixed architecture or specific set of tasks. Finally, our work harnesses efficient triplane 3D representations from (Chan et al., 2022) that is originally designed to generate high-quality, 3D-aware representations from a collection of single-view images. Our method alternatively focuses on the joint learning of dense vision problems and leverages 3D understanding to bring a beneficial structure to the learned representations.

## 3 METHOD

We next briefly review the problem settings for MTL and neural rendering to provide required background and then proceed to describe our proposed method.

### 3.1 MULTI-TASK LEARNING

Our goal is to learn a model $\hat{y}$ that takes in an RGB image $\boldsymbol{I}$ as input and jointly predicts ground-truth labels $Y = \{\boldsymbol{y}_1, \ldots, \boldsymbol{y}_T\}$ for $T$ tasks. In this paper, we focus on dense prediction problems such as semantic segmentation, depth estimation where input image and labels have the same dimensionality. While it is possible to learn an independent model for each task, a more efficient design involves sharing a large portion of the computation across the tasks, via a common feature encoder $f$. Encoder $f$ takes in an image as input and outputs a high-dimensional feature map which has smaller width and height than the input. In this setting, the encoder is followed by multiple task-specific decoders $h_t$ that each ingests $f(\boldsymbol{I})$ to predict corresponding task labels $i.e.$, $h_t(f(\boldsymbol{I}))$, as depicted in Fig. 1 (a).

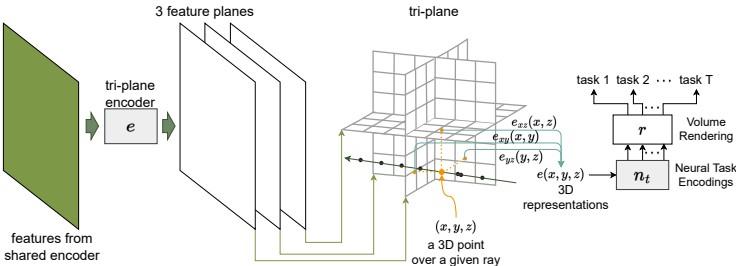

Figure 2: Diagram of 3D-aware regularizer $g$. The regularizer $g$ takes as input the features from the shared encoder and transforms it to a tri-plane using a tri-plane encoder $e$. Given a 3D point $(x, y, z)$ on a given ray, we project the coordinates onto three planes and aggregate features from three planes using summation to obtain the 3D representations, which are then fed into a light-weight MLP $n_t$ to estimate predictions of each task or the density of the 3D point. Finally, in volume rendering $r$, we integrate the predictions over the ray to render the predictions of each task.

Given a labeled training set $\mathcal{D}$ with $N$ image-label pairs, the model weights can be optimized as:

$$\min_{f, \{h_t\}_{t=1}^T} \frac{1}{N} \sum_{(\boldsymbol{I}, Y) \in \mathcal{D}} \sum_{\boldsymbol{y}_t \in Y} \mathcal{L}_t(h_t \circ f(\boldsymbol{I}), \boldsymbol{y}_t), \tag{1}$$

where $\mathcal{L}_t$ is the loss function for task $t$, *i.e.*, cross entropy loss for semantic segmentation, $L_1$ loss for depth estimation. We provide more details in Sec. 4,

## 3.2 3D-Aware Multi-task Learning

An ideal feature extractor $f$ is expected to extract both task-agnostic and task-specific information, towards enabling the following task-specific decoders to solve their respective target tasks accurately. However, in practice, the combination of high-dimensional feature space and highly non-linear mappings from input to output is prone to overfitting to data and learning of noisy correlations. To mitigate these issues, we propose a new 3D-aware regularization technique that first maps extracted features to 3D neural codes, projects them to task-specific fields and finally renders them to obtain predictions for each target task through differentiable rendering (see Fig. 2). In the regularization, outputs for all tasks are conditioned on observations that lie on a low-D manifold (the density (Mildenhall et al., 2020)), enforcing 3D consistency between tasks.

**3D representations.** Training the recent MTL models (*e.g.* Vandenhende et al. (2020b); Ye & Xu (2022)) on high resolution input images for multiple dense prediction tasks simultaneously is computation and memory intensive. Hence, naively mapping their multi-scale high-resolution features to 3D is not feasible due to memory limitations in many standard GPUs. Hence, we adopt the hybrid explicit-implicit tri-plane representations of Chan et al. (2022). In particular, we first feed $\boldsymbol{I}$ into a shared encoder and obtain a $W \times H \times C$-dimensional feature map where $H$ and $W$ are the height and width. Then, through a tri-plane encoder $e$, we project the feature map to three explicit $W \times H \times C'$ dimensional feature maps, $e_{xy}, e_{yz}, e_{xz}$, that represent axis aligned orthogonal feature planes. We can query any 3D coordinate $(x, y, z)$ by projecting it onto each plane, then retrieve the respective features from three planes via bi-linear interpolation and finally aggregate features using summation to obtain the 3D representation ($e(x, y, z) = e_{xy}(x, y) + e_{yz}(y, z) + e_{xz}(x, z)$) as in Chan et al. (2022).

**Neural task fields.** For each task, we use an additional light-weight network $n_t$, implemented as a small MLP, to estimate both a density value and task-specific vector, where this element pair can be denoted as a neural task field for the aggregated 3D representation. We are then able to render these quantities via neural volume rendering Max (1995); Mildenhall et al. (2020) through a differentiable renderer $r$ to obtain predictions for each task. In particular, for the tasks including semantic segmentation, part segmentation, surface normal estimation, boundary detection, saliency prediction, we estimate prediction for each point of a given ray (*e.g.* logits for segmentation) and integrate them over the ray. We normalize the predictions after rendering for surface normal and apply softmax after rendering for segmentation tasks. For depth estimation task, we use the raw prediction as depth maps.

The sequence of mappings can be summarized as: firstly mapping the shared feature encoding $f(\boldsymbol{I})$ to tri-plane features through $e$, further mapping it to neural task fields through $n_t$, finally rendering these to obtain predictions for task $t$, *i.e.* $g_t \circ f(\boldsymbol{I})$ where $g_t = r \circ n_t \circ e$ is the regularizer for task $t$.

**Discussion.** While novel view synthesis methods such as NeRF require the presence of multiple views and knowledge of the camera matrices, here we assume a single view to extract the corresponding 3D representations and to render them as task predictions. For rendering, we assume that the camera is orthogonal to image center here, and depict $r$ as a function that takes only the output of $n_t$ but not the viewpoint as input. In the experiments, we show that our model consistently improves the MTL performance, even when learned from a single view per scene, thanks to the 3D structure of representations imposed by our regularizer.

**Optimization.** We measure the mismatch between ground-truth labels and the predictions obtained from our 3D-aware model branch, and use this signal to jointly optimize the model along with the original common task losses found in Eq. (1):

$$\min_{f, \{h_t, g_t\}_{t=1}^{T}} \frac{1}{N} \sum_{(\boldsymbol{I}, Y) \in \mathcal{D}} \sum_{\boldsymbol{y}_t \in Y} \mathcal{L}_t(h_t \circ f(\boldsymbol{I}), \boldsymbol{y}_t) + \alpha_t \underbrace{\mathcal{L}_t(g_t \circ f(\boldsymbol{I}), \boldsymbol{y}_t)}_{\text{3D-aware regularizer}}, \qquad (2)$$

where $\alpha_t$ is a hyperparameter balancing loss terms.

**Cross-view consistency.** Though our 3D-aware regularizer does not require multiple views of the same scene to be presented, it can be easily extended to penalize the cross-view inconsistency on the predictions when multiple views of the same scene are available, *e.g.* video frames. Given two views of a scene $\boldsymbol{I}$ and $\boldsymbol{I}'$ with their camera respective viewpoints $V$ and $V'$, we compute predictions for $\boldsymbol{I}'$ but by using $\boldsymbol{I}$ as the input and render it by using the relative camera transformation $\Delta V$ from $V$ to $V'$. Then we further penalize the inconsistency between this prediction and ground-truth labels of $\boldsymbol{I}'$:

$$\min_{f, \{h_t, g_t\}_{t=1}^{T}} \frac{1}{N} \sum_{\substack{\{(\boldsymbol{I}, Y), \\ (\boldsymbol{I}', Y')\} \in \mathcal{D}}} \sum_{\substack{\boldsymbol{y}_t \in Y, \\ \boldsymbol{y}_t' \in Y'}} \mathcal{L}_t(h_t \circ f(\boldsymbol{I}), \boldsymbol{y}_t) + \alpha_t \underbrace{\mathcal{L}_t(g_t \circ f(\boldsymbol{I}), \boldsymbol{y}_t)}_{\text{3D-aware regularizer}} + \alpha_t' \underbrace{\mathcal{L}_t(g_t^{\Delta V} \circ f(\boldsymbol{I}), \boldsymbol{y}_t')}_{\text{cross-view regularizer}},$$
$$(3)$$

where $\alpha_t$ and $\alpha_t'$ are hyperparameters balancing loss terms. We set $\alpha_t = \alpha_t'$. Note that in this case $g_t^{\Delta V}$ is a function of $\Delta V$, as the relative viewpoint $\Delta V$ is used by the renderer $r$.

## 4 EXPERIMENTS

Here we first describe the benchmarks used and our implementation details, then present a quantitative and qualitative analysis of our method.

### 4.1 DATASET

**NYUv2 (Silberman et al., 2012):** It contains 1449 RGB-D images, sampled from video sequences from a variety of indoor scenes, which we use to perform four tasks; namely 40-class semantic segmentation, depth estimation, surface normal estimation and boundary detection in common with prior work (Ye & Xu, 2022; Bruggemann et al., 2021). Following the previous studies, we use the true depth data recorded by the Microsoft Kinect and surface normals provided in the prior work (Eigen & Fergus, 2015) for depth estimation and surface normal estimation tasks.

**NYUv2 video frames:** In addition to the standard data split, NYUv2 (Silberman et al., 2012) also provides additional video frames[1] which are labeled only for depth estimation. Only for the cross-view consistent regularization experiments, we merge the original split with video frames, and train MTL models by minimizing loss on available labeled tasks, *i.e.* all four tasks on the original data and only the depth on video frames. To estimate the relative camera pose $\Delta V$ between the frames, we use COLMAP (Schönberger & Frahm, 2016; Schönberger et al., 2016).

---

[1]`https://www.kaggle.com/datasets/soumikrakshit/nyu-depth-v2`

**PASCAL-Context (Chen et al., 2014):** PASCAL (Everingham et al., 2010) is a commonly used image benchmark for dense prediction tasks. We use the data splits from PASCAL-Context (Chen et al., 2014) which has annotations for semantic segmentation, human part segmentation and semantic edge detection. Additionaly, following (Vandenhende et al., 2021; Ye & Xu, 2022), we also consider surface normal prediction and saliency detection using the annotations provided by Vandenhende et al. (2021).

## 4.2 IMPLEMENTATION DETAILS

Our regularizer is architecture agnostic and can be applied to different architectures. In our experiments, it is incorporated into two state-of-the-art (SotA) MTL methods; MTI-Net (Vandenhende et al., 2020b) and InvPT (Ye & Xu, 2022) which builds on the convolutional neural network (CNN), HRNet-48 (Wang et al., 2020) and transformer based ViT-L (Dosovitskiy et al., 2020) respectively. In all experiments, we follow identical training, evaluation protocols (Ye & Xu, 2022). We append our 3D-aware regularizer to these two models using two convolutional layers, followed by BatchNorm and ReLU, to project feature maps to the tri-plane space resulting in a common size and channel width (64). A 2-layer MLP is used to render each task as in Chan et al. (2022). We use identical hyper-parameters; learning rate, batch size, loss weights, loss functions, pre-trained weights, optimizer, evaluation metrics as MTI-Net and InvPT, respectively. We jointly optimize the task-specific losses and losses arising from our 3D regularization. During inference, the regularizer is discarded. We refer to the supplementary material for further details.

## 4.3 RESULTS

**Comparison with SotA methods.** We compare our method with the SotA MTL methods on NYUv2 and PASCAL-Context datasets and report results in Tab. 1 and Tab. 2, respectively. Following Bruggemann et al. (2021), we use HRNet-48 (Wang et al., 2020) as backbone when comparing to CNN based methods; Cross-Stitch (Misra et al., 2016), PAP (Zhang et al., 2019), PSD (Zhou et al., 2020), PAD-Net (Xu et al., 2018), ATRC (Bruggemann et al., 2021), MTI-Net (Vandenhende et al., 2020b). We use ViT-L (Dosovitskiy et al., 2020) as backbone when comparing to InvPT (Ye & Xu, 2022).

In NYUv2 (see Tab. 1), when using HRNet-48 as backbone, we observe that ATRC (Bruggemann et al., 2021) and MTI-Net (Vandenhende et al., 2020b) obtain the best performance. By incorporating our method to MTI-Net (Vandenhende et al., 2020b), we improve its performance on all tasks and outperform all CNN based MTL methods. In comparison, the InvPT approach (Ye & Xu, 2022) achieves superior MTL performance by leveraging both the ViT-L (Dosovitskiy et al., 2020) backbone and multi-scale cross-task interaction modules. Our method is also able to quantitatively improve upon the base InvPT by integrating our proposed 3D-aware regularizer, *e.g.* +1.31 mIoU on Seg. The results evidence that the geometric information is beneficial for jointly learning multiple dense prediction tasks.

| Method | Backbone | Seg. (mIoU) ↑ | Depth (RMSE) ↓ | Normal (mErr) ↓ | Boundary (odsF) ↑ |
|---|---|---|---|---|---|
| Cross-Stitch (Misra et al., 2016) | HRNet-48 | 36.34 | 0.6290 | 20.88 | 76.38 |
| PAP (Zhang et al., 2019) | HRNet-48 | 36.72 | 0.6178 | 20.82 | 76.42 |
| PSD (Zhou et al., 2020) | HRNet-48 | 36.69 | 0.6246 | 20.87 | 76.42 |
| PAD-Net (Xu et al., 2018) | HRNet-48 | 36.61 | 0.6270 | 20.85 | 76.38 |
| ATRC (Bruggemann et al., 2021) | HRNet-48 | 46.33 | 0.5363 | 20.18 | 77.94 |
| MTI-Net (Vandenhende et al., 2020b) | HRNet-48 | 45.97 | 0.5365 | 20.27 | 77.86 |
| Ours | HRNet-48 | **46.67** | **0.5210** | **19.93** | **78.10** |
| InvPT (Ye & Xu, 2022) | ViT-L | 53.56 | 0.5183 | 19.04 | 78.10 |
| Ours | ViT-L | **54.87** | **0.5006** | **18.55** | **78.30** |

Table 1: Quantitative comparison of our method to the SotA methods; NYUv2 dataset.

Tab. 2 depicts experimental results on the PASCAL-Context dataset where previous method results are reproduced from Ye & Xu (2022). We also report results from our implementation of the MTI-Net, denoted by 'MTI-Net*', where we found that our implementation obtains better performance. We observe that the performance of existing methods is better than in the previous NYUv2 experiment (Tab. 1), as PASCAL-Context has significantly more images available for training. From Tab. 2 we observe that our method, incorporating our proposed regularizer to MTI-Net (Vandenhende et al., 2020b), can improve the performance on all tasks with respect to our base MTI-Net implementation, *e.g.* +2.29 mIoU on Seg, and obtains the best performance on most tasks compared with MTL

methods that use the HRNet-48 backbone. As in NYUv2, the InvPT model (Ye & Xu, 2022) achieves better performance on a majority of tasks over existing methods. Our method with InvPT again obtains improvements on all tasks over InvPT, *e.g.* +1.51 mIoU on PartSeg and +1.00 odsF on Boundary. This result further suggests that our method is effective for enabling the MTL network to learn beneficial geometric cues and that the technique can be incorporated with various MTL methods for comprehensive task performance improvements.

| Method | Backbone | Seg. (mIoU) ↑ | PartSeg (mIoU) ↑ | Sal (maxF) ↑ | Normal (mErr) ↓ | Boundary (odsF) ↑ |
|---|---|---|---|---|---|---|
| ASTMT (Maninis et al., 2019) | HRNet-48 | 68.00 | 61.10 | 65.70 | 14.70 | 72.40 |
| PAD-Net (Xu et al., 2018) | HRNet-48 | 53.60 | 59.60 | 65.80 | 15.30 | 72.50 |
| MTI-Net (Vandenhende et al., 2020b) | HRNet-48 | 61.70 | 60.18 | 84.78 | 14.23 | 70.80 |
| ATRC (Bruggemann et al., 2021) | HRNet-48 | 62.69 | 59.42 | 84.70 | 14.20 | 70.96 |
| MTI-Net* (Vandenhende et al., 2020b) | HRNet-48 | 64.42 | 64.97 | 84.56 | 13.82 | 74.30 |
| Ours | HRNet-48 | **66.71** | **65.20** | 84.59 | **13.71** | **74.50** |
| InvPT (Ye & Xu, 2022) | ViT-L | 79.03 | 67.61 | 84.81 | 14.15 | 73.00 |
| Ours | ViT-L | **79.53** | **69.12** | **84.94** | **13.53** | **74.00** |

Table 2: Quantitative comparison of our method to the SotA methods; PASCAL-Context dataset.

**3D regularizer with multiple views.** Here we investigate whether learning stronger 3D consistency across multiple views with our regularizer further improves the performance in multiple tasks. To this end, we merge the NYUv2 dataset with the additional video frames possessing only depth annotation and train the base InvPT, our method and our method with cross-view consistency on the merged data. For InvPT, we train the model by minimizing losses over the labeled tasks. We train our method by minimizing both the supervised losses and the 3D-aware regularization loss. We further include the cross-view consistency loss as in Eq. (3).

| Method | Seg. (mIoU) ↑ | Depth (RMSE) ↓ | Normal (mErr) ↓ | Boundary (odsF) ↑ |
|---|---|---|---|---|
| InvPT (Ye & Xu, 2022). | 53.44 | 0.4927 | 18.78 | 77.90 |
| Ours | 54.93 | 0.4879 | **18.47** | 77.90 |
| Ours with cross-view consistency | **54.99** | **0.4850** | 18.52 | **78.00** |

Table 3: Quantitative comparison of our method on NYUv2 dataset + extra video frames with multiple views.

Results of the three approaches are reported in Tab. 3. Compared with the results in Tab. 1, we can see that including video frames for training improves the performance of InvPT on depth and surface normal tasks while yielding comparable performance on remaining tasks. We also see that our method obtains consistent improvement over the InvPT on four tasks with applying 3D-aware regularization using only a single view. Adding the cross-view consistency loss term to our method, we can observe further performance improvement beyond using only single view samples. This suggests that better 3D geometry learning through multi-view consistency is beneficial, however, the improvements are modest. We argue that coarse 3D scene information obtained from single views can be sufficient to learn more structured and regulate inter-task relations.

We also note that this experimental setting is also related to the recent MTL work (Li et al., 2022) that can learn from partially annotated data by exploiting cross-task relations. However we here focus on an orthogonal direction and believe our complementary works have scope to be integrated together. We leave this as a promising direction for future work.

**Comparison with auxiliary network heads.** Prior work suggests that the addition of auxiliary heads performing the same task with identical head architectures yet with different weight initializations can be further helpful to performance (Meyerson & Miikkulainen, 2018). To verify whether the improvements obtained by our regularizer is not due to the additional heads solely but introduced 3D structure, we conduct a comparison with our baseline and report results in Tab. 4. The results show that adding auxiliary heads ('InvPT + Aux. Heads') does not necessarily lead to better performance on all tasks; *e.g.* Seg, whereas our method can be seen to outperform this baseline on all tasks suggesting the benefit of introducing 3D-aware structure across tasks.

| Method | Seg. (mIoU) ↑ | Depth (RMSE) ↓ | Normal (mErr) ↓ | Boundary (odsF) ↑ |
|---|---|---|---|---|
| InvPT (Ye & Xu, 2022) | 53.56 | 0.5183 | 19.04 | 78.10 |
| InvPT + Aux. Heads | 52.45 | 0.5131 | 18.90 | 77.60 |
| Ours | **54.87** | **0.5006** | **18.55** | **78.30** |

Table 4: Quantitative comparison of our method to the baseline of adding auxiliary heads to InvPT; NYUv2 dataset.

**3D-aware regularizer predictions.** Though we discard the regularizer during inference, the regularizer can also be used to produce predictions for the tasks. To investigate their estimation utility, we report task performance using the default task specific heads $h_t$, the regularizer output (*regularizer*) and finally using the averaged predictions over two in Tab. 5. We observe that the regularizer alone estimations are worse than the task-specific heads, however, the performance of their averaged output yields marginal improvements to the boundary detection task. The lower performance of using the regularizer alone may be explained by the fact that the rendering image size is typically small (*e.g.* we render 56×72 images for NYUv2). The addition of a super-resolution module, similar to previous work (Chan et al., 2022), can further improve the quality of the related predictions. We leave this to future work.

| outputs | Seg. (mIoU) ↑ | Depth (RMSE) ↓ | Normal (mErr) ↓ | Boundary (odsF) ↑ |
|---|---|---|---|---|
| task-specific heads | **54.87** | **0.5006** | **18.55** | 78.30 |
| regularizer | 51.79 | 0.5282 | 18.90 | 74.80 |
| avg | 54.68 | 0.5062 | 18.70 | **78.50** |

Table 5: Quantitative results of the predictions from the task-specific heads, regularizer or the average of both the task-specific heads and regularizer in our method; NYUv2 dataset.

**Tasks for 3D-aware regularizer.** Our regularizer renders predictions for all learning tasks by default. We further study the effect of isolating different tasks for rendering with the regularizer in Tab. 6. Specifically; we jointly optimize the MTL network with a regularizer that renders only one individual task predictions. From Tab. 6 we observe that rendering different individual tasks in the regularizer leads to only marginally differing results and yet using all tasks for rendering can help to better learn the geometric information for MTL, *i.e.* 'All tasks' obtains the best performance on the majority of tasks.

| render tasks | Seg. (mIoU) ↑ | Depth (RMSE) ↓ | Normal (mErr) ↓ | Boundary (odsF) ↑ |
|---|---|---|---|---|
| Seg. | **55.04** | 0.5038 | 18.78 | 77.90 |
| Depth | 54.62 | 0.5041 | 18.93 | 77.50 |
| Normal | 53.43 | 0.5117 | 18.59 | 77.50 |
| Edge | 53.97 | 0.5022 | 18.97 | 77.50 |
| All tasks | 54.87 | **0.5006** | **18.55** | **78.30** |

Table 6: Quantitative results of our method isolating different tasks for rendering with the regularizer; NYUv2 dataset.

**Using less data.** We further investigate the performance gain obtained by our method when trained with fewer training samples. To this end, we train the baseline InvPT (Ye & Xu, 2022) and our method on 25% and 50% of the NYUv2 data after randomly subsampling the original training set. The results are reported in Tab. 7. As expected, more training samples result in better performance in all cases. Our method consistently outperforms the baseline on all tasks in all label regimes with higher margins when more data is available. As the full NYUv2 training set is relatively small, contains only 795 images, our regularizer learns better 3D consistency across tasks from more data too, hence resulting enhanced task performance.

| # images | Method | Seg. (mIoU) ↑ | Depth (RMSE) ↓ | Normal (mErr) ↓ | Boundary (odsF) ↑ |
|---|---|---|---|---|---|
| 795 (100%) | InvPT (Ye & Xu, 2022) | 53.56 | 0.5183 | 19.04 | 78.10 |
| | Ours | **54.87** | **0.5006** | **18.55** | **78.30** |
| 397 (50%) | InvPT (Ye & Xu, 2022) | 49.24 | 0.5741 | 20.60 | 74.90 |
| | Ours | **49.30** | **0.5656** | **20.30** | **76.50** |
| 198 (25%) | InvPT (Ye & Xu, 2022) | 43.83 | 0.6060 | 21.76 | 74.80 |
| | Ours | **44.79** | **0.5972** | **21.57** | 74.80 |

Table 7: Quantitative comparison of the baseline InvPT and our method in the NYUv2 dataset for varying training set sizes.

## 4.4 QUALITATIVE RESULTS

We visualize the task predictions for both our method and the base InvPT method on an NYUv2 sample in Fig. 3. Our method can be observed to estimate better predictions consistently for four tasks. For example, our method estimates more accurate predictions around the boundary of the refrigerator, stove and less noisy predictions within objects like curtain and stove. The geometric information learned in our method helps distinguish different adjacent objects, avoids noisy predictions within

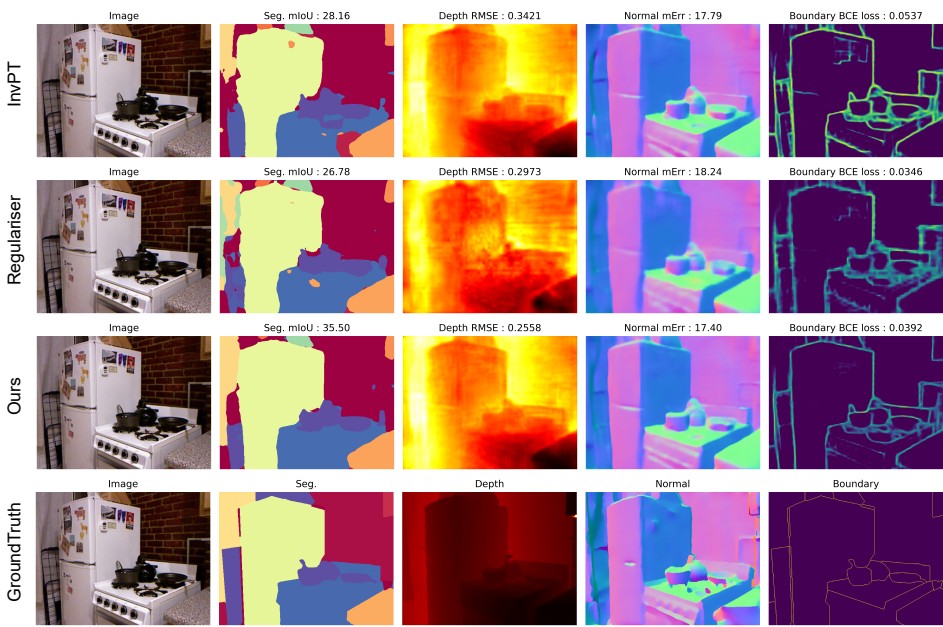

Figure 3: Qualitative results on NYUv2. Each column shows the image or predictions and performance for each task. The last row shows the ground-truth of four tasks. The first to the third row shows the predictions of InvPT, the regularizer in our method and task-specific decoders of our method, respectively.

object boundaries and also improves the consistency across tasks as in the regularizer, all tasks predictions are rendered based on the same density.

We then visualize the predictions of our method's regularizer and the task-specific decoder NYUv2 in Fig. 3. As shown in the figure, our regularizer can also render high quality predictions for different tasks yet it was observed to obtain worse quantitative performance than the task-specific decoders. As discussed, this is due to the rendering image size being usually small (*e.g.* we render $56 \times 72$ images for NYUv2).

## 5 CONCLUSION AND LIMITATIONS

We demonstrate that encouraging 3D-aware interfaces between different related tasks including depth estimation, semantic segmentation and surface normal estimation consistently improves the multitask performance when incorporated to the recent MTL techniques in two standard dense prediction benchmarks. Our model can be successfully used with different backbone architectures and does not bring any additional inference costs. Our method has limitations too. Despite the efficient 3D modeling through the triplane encodings, representing 3D representations for higher resolution 3D volumes is still expensive in terms of memory or computational cost. Rendering specular objects will require different rendering or objects with high frequency 3D details may require more accurate 3D modeling. Furthermore, we balance loss functions with fixed cross-validated hyperparameters, while it would be more beneficial to use adaptive loss balancing strategies (Kendall et al., 2018) or discarding conflicting gradients (Liu et al., 2021a). Finally, in the cross-view consistency experiments where only a portion of images are labeled for all the tasks, our method does not make use of semi-supervised learning or view-consistency for the tasks with missing labels which can further improve the performance of our model.

**Acknowledgement.** HB is supported by the EPSRC programme grant Visual AI EP/T028572/1. We thank Octave Mariotti, Changjian Li, and Titas Anciukevicius for their valuable feedback.

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

## A IMPLEMENTATION DETAILS

We implement our approach in conjunction with state-of-the-art multi-task learning methods; MTI-Net (Vandenhende et al., 2020b) and InvPT (Ye & Xu, 2022) while following identical training, evaluation protocols (Ye & Xu, 2022). We use HRNet-48 (Wang et al., 2020) and ViT-L (Dosovitskiy et al., 2020) to serve as shared encoders and append our 3D-aware regularizer to MTI-Net and InvPT using two convolutional layers, followed by BatchNorm, ReLU, and dropout layer with a dropout rate of 0.15 to transform feature maps to the tri-plane dimensionality, resulting in a common size and channel width ($64$). A 2-layer MLP with 64 hidden units as in Chan et al. (2022) and a LeakyReLU non-linearity with the negative slope of -0.2 as in Skorokhodov et al. (2022), is used to render each task as in Chan et al. (2022). We use identical hyper-parameters; learning rate, batch size, loss weights, loss functions, pre-trained weights, optimizer, evaluation metrics as MTI-Net and InvPT, respectively. We jointly optimize task-specific losses and losses arising from our 3D regularization. During inference, the regularizer is discarded. We use the same task-specific loss weights as in Ye & Xu (2022). We train all models for 40K iterations with a batch size of 6 for experiments of using InvPT as in Ye & Xu (2022) and a batch size of 8 for experiments of using MTI-Net as in (Vandenhende et al., 2020b). We ramp up the $\alpha_t$ from 0 to 4 linearly in 20K iterations and keep $\alpha_t = 4$ for the rest 20K iterations. In the regularizer, we assume that the camera is orthogonal to image center, and depict $r$ as a function that takes only the output of $n_t$ but not the viewpoint as input. In a 3D coordinates $(x, y, z)$, the $x$ and $y$ coordinates are aligned with pixel locations and $z$ is the depth value. We further use a two-pass importance sampling as in NeRF (Mildenhall et al., 2020). For the majority of the experiments in the manuscript, we use 128 total depth samples per ray. We render $56 \times 72$ predictions for NYUv2 and $64 \times 64$ for PASCAL-Context and resize the predictions via bilinear interpolation to the groundtruth resolution. Our code and models will be made public based upon acceptance.

## B TRAINING COST ANALYSIS

| Method | Time | Memory | Params. | FLOPS |
|---|---|---|---|---|
| MTI-Net (Vandenhende et al., 2020b) | 1.000 | 1.000 | 1.000 | 1.000 |
| Ours | 1.489 | 1.638 | 1.005 | 1.263 |
| InvPT (Ye & Xu, 2022) | 1.000 | 1.000 | 1.000 | 1.000 |
| Ours | 1.318 | 1.397 | 1.016 | 1.114 |

Table 8: Training Cost Comparisons to MTI-Net and InvPT; NYUv2 dataset. Note that our method has no additional inference cost as the regularizer is discarded during testing.

## C RESULTS OVER MULTIPLE RUNS

Here, we report the results of our method over 3 runs on NYUv2 and PASCAL-Context and report the results in Tabs. 9 and 10. From the results, we can see that our method is stable (i.e. the std is very small on each task) and improves over the baseline consistently on all tasks.

| Method | Seg. (mIoU) ↑ | Depth (RMSE) ↓ | Normal (mErr) ↓ | Boundary (odsF) ↑ |
|---|---|---|---|---|
| InvPT (Ye & Xu, 2022) | 53.56 | 0.5183 | 19.04 | 78.10 |
| Ours | **54.86 ± 0.29** | **0.5000 ± 0.0010** | **18.49 ± 0.09** | **78.17 ± 0.09** |

Table 9: Quantitative comparison of our method to the InvPT over 3 runs; NYUv2 dataset.

| Method | Seg. (mIoU) ↑ | PartSeg (mIoU) ↑ | Sal (maxF) ↑ | Normal (mErr) ↓ | Boundary (odsF) ↑ |
|---|---|---|---|---|---|
| InvPT (Ye & Xu, 2022) | 79.03 | 67.61 | 84.81 | 14.15 | 73.00 |
| Ours | **79.92 ± 0.32** | **69.08 ± 0.15** | **84.85 ± 0.06** | **13.70 ± 0.14** | **73.83 ± 0.17** |

Table 10: Quantitative comparison of our method to the InvPT over 3 runs; PASCAL-Context dataset.

Here, we analyze memory and computational cost during training for tackling four tasks in NYUv2 and report them in Tab. 8. As shown in Tab. 8, our method that incorporates the regularizer to the MTL baseline slightly increases the number of parameters (Ours vs InvPT: 1.016 vs 1) and FLOPS (Ours vs InvPT: 1.114 vs 1) during training, training time (Ours vs InvPT: 1.318 vs 1), and training memory

(Ours vs InvPT: 1.397 vs 1). We highlight that there is \*NO additional cost\* during inference, since the regularizer will be discarded during inference.

## D    COMPARISONS WITH MORE RECENT SOTA

| Method | Seg. (mIoU) ↑ | Depth (RMSE) ↓ | Normal (mErr) ↓ | Boundary (odsF) ↑ |
|---|---|---|---|---|
| TaskPromper (Ye & Xu, 2023a) | 55.30 | 0.5152 | **18.47** | 78.20 |
| TaskExpert (Ye & Xu, 2023b) | **55.35** | 0.5157 | 18.54 | **78.40** |
| InvPT (Ye & Xu, 2022) | 53.56 | 0.5183 | 19.04 | 78.10 |
| Ours | 54.87 | **0.5006** | 18.55 | 78.30 |

Table 11: Quantitative comparison of our method to more SotA methods; NYUv2 dataset.

| Method | Seg. (mIoU) ↑ | PartSeg (mIoU) ↑ | Sal (maxF) ↑ | Normal (mErr) ↓ | Boundary (odsF) ↑ |
|---|---|---|---|---|---|
| TaskPrompter (Ye & Xu, 2023a) | **80.89** | 68.89 | 84.83 | 13.72 | 73.50 |
| TaskExpert (Ye & Xu, 2023b) | 80.64 | **69.42** | 84.87 | 13.56 | 73.30 |
| InvPT (Ye & Xu, 2022) | 79.03 | 67.61 | 84.81 | 14.15 | 73.00 |
| Ours | 79.53 | 69.12 | **84.94** | **13.53** | **74.00** |

Table 12: Quantitative comparison of our method to the SotA methods; PASCAL-Context dataset.

We include the comparisons of our method incorporated with InvPT to more recent state-of-the-art methods, including TaskPrompter (Ye & Xu, 2023a) and TaskExpert (Ye & Xu, 2023b) and report the results in Tabs. 11 and 12. Methods from Liu et al. (2023) and Chen et al. (2023) are not compared as they did not reported results on NYUv2 and PASCAL benchmarks with the same backbone. Note that TaskExpert (Ye et al., 2023b) is published after we submitting the manuscript. From the results shown in Tab. 11, we can see that, our method incorporated with InvPT achieves much better result on Depth while comparable results on the rest of tasks in NYUv2 compared with TaskPrompter and TaskExpert. In PASCAL benchmark, from Tab. 12 we can see that our method obtains much better results on saliency, surface normal and boundary estimation while obtaining comparable result on Human part segmentation and slightly worse on semantic segmentation. TaskPrompter adds learnable prompts for refining the feautures and the TaskExpert emsembles task-specific features from multiple task-specific experts for final task predictions and they all increase the capacity of the network to achieve better results. Also, they can potentially be complementary to our method and we believe incorporating our method with them can further improve the performance in multi-task learning by regulating the shared features to be 3D-aware with no additional cost during inference.

## E    DISCUSSION

**Camera parameters.**    In our paper, the 3D coordinates and strategy of projecting the 3D coordinates onto the feature planes are similar to the ones in PiFU (Saito et al., 2019) and (Yao et al., 2023). The feature planes are generated by the feature encoder and it is pixel-wise feature map instead of a global pooled feature vector. The $x$ and $y$ coordinates are aligned with pixel locations and $z$ is the depth value. We follow Chan et al. (2022) that projects the coordinates $(x, y, z)$ onto three planes $e_{xy}, e_{yz}, e_{xz}$, retrieving the features via bilinear interpolation, and aggregates features from three planes instead of taking the 2D features and the $z$ values as representations in PiFU (Saito et al., 2019) or dividing the dimension of the feature map channel into $D$ groups ($D$ is the number of depth bins) in (Yao et al., 2023). So our method has similar property as in PiFU (Saito et al., 2019) and (Yao et al., 2023) and does not overfit to the camera parameters.

Also, as we first feed the image into the feature encoder, which should be scaling the 3D coordinates accordingly and the coordinates will not be absolute but at the right scale for rendering. After training, the 3D-aware regularizer is discarded and we only use the multi-task learning branch for generating predictions for different tasks. We also visualize multiple images' predictions of the regularizers on PASCAL in Fig. 4. The PASCAL dataset consists of annotated consumer photographs collected from the flickr photo-sharing web-site, taken by various cameras with different intrinsics. From Fig. 4, we can see that the regularizer can render good quality predictions for all tasks on all images which also indicates that it does not overfit to the camera parameters.

**Limitations and future work.**    Despite the efficient 3D modeling through the triplane encodings, representing 3D representations for higher resolution 3D volumes is still expensive in terms of

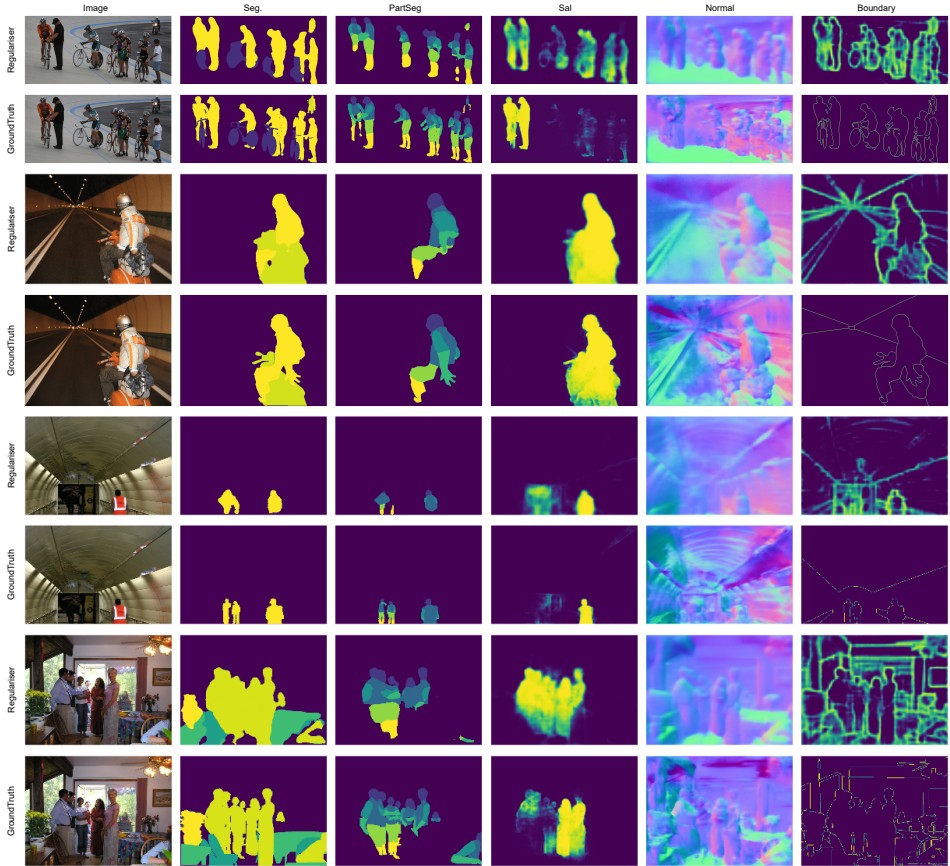

Figure 4: Qualitative results on PASCAL. Each column shows the image or predictions of our method's regularizer branch or the groundtruth for each task, respectively.

memory or computational cost. Some common efficient sampling strategies such as random sampling and pixel binning can be useful for reducing the cost. The tri-plane generated from the feature encoder can be relatively small resolution due to the feature downsampling and requires upsampling strategies for generating higher resolution feature planes for better rendering while it will inevitably increase the training cost. Additionally, rendering specular objects will require different rendering or objects with high frequency 3D details may require more accurate 3D modeling. Though our proposed method obtains performance gains consistently over multiple tasks, we balance loss functions with fixed cross-validated hyperparameters, while it would be more beneficial to use adaptive loss balancing strategies (Kendall et al., 2018) or discarding conflicting gradients (Liu et al., 2021a). Finally, in the cross-view consistency experiments where only some of the images are labeled for all the tasks, our method does not make use of semi-supervised learning or view-consistency for the tasks with missing labels which can be further improve the performance of our model. We believe that more advanced techniques in 3D modeling can further improve our method for rendering higher resolution predictions with higher efficiency and better regulating the cross-task correlations and cross-view consistency.

