# OpenReview forum: "Multi-task Learning with 3D-Aware Regularization"
_ICLR.cc/2024/Conference — ICLR 2024 poster_

### Official Review · Reviewer_4RSv · 2023-10-28

**Soundness:** 3 good
**Presentation:** 3 good
**Contribution:** 3 good
**Rating:** 6
**Confidence:** 5

**Summary:**

The paper proposes a structured 3D-aware regularizer for multi-task learning (MTL) in computer vision. The regularizer interfaces multiple tasks by projecting features from an image encoder to a shared 3D feature space and decoding them into task output space through differentiable rendering.

**Strengths:**

* The paper provides clear explanations of the proposed method, the problem it addresses, and the evaluation process.

* The paper's contributions have significant implications for multi-task learning in computer vision. The structured 3D-aware regularizer can be integrated into existing models, improving their performance and reducing noise in cross-task correlations.

**Weaknesses:**

* The paper could benefit from a more extensive analysis of the proposed method's computational efficiency. While it is mentioned that the regularizer does not introduce additional computational cost for inference, a more detailed analysis or comparison with other methods in terms of computational efficiency would strengthen the paper.

* It would be beneficial to discuss the limitations or potential failure cases of the proposed 3D-aware regularizer. It would be valuable to address any potential drawbacks or scenarios where the regularizer may not be as effective.

* The paper lacks a thorough comparison with existing state-of-the-art methods in multi-task learning. It would be beneficial to include comparisons with other regularization techniques or approaches that address the problem of noisy cross-task correlations.

**Questions:**

See weakness section for more details.

---

> ### Author Response · Authors · 2023-11-21
> **Response to Reviewer 4RSv**
>
> We thank the reviewer for the positive feedback and insightful comments. We address the questions and comments below.
>
> > Q1: The paper could benefit from a more extensive analysis of the proposed method's computational efficiency. While it is mentioned that the regularizer does not introduce additional computational cost for inference, a more detailed analysis or comparison with other methods in terms of computational efficiency would strengthen the paper.
>
> R1: Thanks for the suggestion. We include a comparison of computational and memory cost during training in the supplementary. Our method that incorporates the regularizer to the MTL baseline slightly increases the number of parameters (Ours vs InvPT: 1.016 vs 1) and FLOPS (Ours vs InvPT: 1.114 vs 1) during training, training time (Ours vs InvPT: 1.318 vs 1), and training memory (Ours vs InvPT: 1.397 vs 1), and has *NO additional cost* during the inference since the regularizer will be discarded during inference.
>
> > Q2: It would be beneficial to discuss the limitations or potential failure cases of the proposed 3D-aware regularizer. It would be valuable to address any potential drawbacks or scenarios where the regularizer may not be as effective.
>
> R2: Thanks for the comment. We included discussion of limitations of the regularizer in the supplementary "Despite the efficient 3D modeling through the triplane encodings, representing 3D representations for higher resolution 3D volumes is still expensive in terms of memory or computational cost. The tri-plane generated from the feature encoder can be relatively small resolution due to the feature downsampling and requires upsampling strategies for generating higher resolution feature planes for better rendering while it will inevitably increase the training cost. Additionally, rendering specular objects will require different rendering or objects with high frequency 3D details may require more accurate 3D modeling. Finally, in the cross-view consistency experiments, where only some of the images are labeled for all the tasks, our method does not make use of semi-supervised learning or view-consistency for the tasks with missing labels which can be further improve the performance of our model."

---

> ### Author Response · Authors · 2023-11-21
> **Response to Reviewer 4RSv**
>
> > Q3: The paper lacks a thorough comparison with existing state-of-the-art methods in multi-task learning. It would be beneficial to include comparisons with other regularization techniques or approaches that address the problem of noisy cross-task correlations.
>
> R3: We compared recent state-of-the-art methods, including the most recent approaches, InvPT and MTI-Net that follow the same strategies i.e. sharing the feature encoder across all tasks. The InvPT has already achieved very competitive performance in multi-task learning. We are awared of more recent methods on multi-task learning by leveraging prompt learning (Ye et al., 2023a, Liu et al., 2023) and mixture-of-expert (MoE) (Chen et al., 2023, Ye et al., 2023b) techniques, which we discussed in the our related work section.
>
> We include the comparisons of our method incorporated with InvPT to the TaskPrompter (Ye et al., 2023a) and TaskExpert (Ye et al., 2023b) in the supplementary and below. Methods from Liu et al., (2023) and Chen et al., (2023) are not compared as they did not reporte results on NYUv2 and PASCAL benchmarks with the same backbone. Note that TaskExpert (Ye et al., 2023b) is published after we submitting the manuscript. From the results shown below, we can see that, our method incorporated with InvPT achieves much better result on Depth while comparable results on the rest of tasks in NYUv2 compared with TaskPrompter and TaskExpert. In PASCAL benchmark, we can see that our method obtains much better results on saliency, surface normal and boundary estimation while obtaining comparable result on Human part segmentation and slightly worse on semantic segmentation. TaskPrompter adds learnable prompts for refining the feautures and the TaskExpert emsembles task-specific features from multiple task-specific experts for final task predictions and they all increase the capacity of the network to achieve better results. Also, they can potentially be complementary to our method and we believe incorporating our method with them can further improve the performance in multi-task learning by regulating the shared features to be 3D-aware with no additional cost during inference.
>
> > Comparisons with more recent SotA methods on NYUv2
> |    Method    | Seg. (mIoU) | Depth (RMSE) | Normal (mErr) | Boundary (odsF) |
> |:------------:|:-----------:|:------------:|:-------------:|:---------------:|
> | TaskPrompter |    55.30    |    0.5152    |   **18.47**   |      78.20      |
> |  TaskExpert  |  **55.35**  |    0.5157    |     18.54     |    **78.40**    |
> |     InvPT    |    53.56    |    0.5183    |     19.04     |      78.10      |
> |     Ours     |    54.87    |  **0.5006**  |     18.55     |      78.30      |
>
> > Comparisons with more recent SotA methods on PASCAL
> |    Method    | Seg. (mIoU) | PartSeg (mIoU) | Sal (maxF) | Normal (mErr) | Boundary (odsF) |
> |:------------:|:-----------:|:--------------:|:----------:|:-------------:|:---------------:|
> | TaskPrompter |  **80.89**  |      68.89     |    84.83   |     13.72     |      73.50      |
> |  TaskExpert  |    80.64    |    **69.42**   |    84.87   |     13.56     |      73.30      |
> |     InvPT    |    79.03    |      67.61     |    84.81   |     14.15     |      73.00      |
> |     Ours     |    79.53    |      69.12     |  **84.94** |   **13.53**   |    **74.00**    |
>
> We compared our method with alternative regularization techniques for multi-task learning in Table 4, where the compared method (Meyerson & Miikkulainen, 2018) learns auxiliary heads for achieving better performance. As shown in Table 4, the results show that adding auxiliary heads (‘InvPT + Aux. Heads’) does not necessarily lead to better performance on all tasks; e.g. Seg, whereas our method can be seen to outperform this baseline on all tasks suggesting the benefit of introducing 3D-aware structure across tasks.
>
> Ye et al., TASKPROMPTER: SPATIAL-CHANNEL MULTI-TASK PROMPTING FOR DENSE SCENE UNDERSTANDING, ICLR 2023a,
>
> Liu et al., Hierarchical Prompt Learning for Multi-Task Learning, CVPR, 2023
>
> Chen et al., Mod-Squad: Designing Mixture of Experts As Modular Multi-Task Learners, CVPR, 2023
>
> Ye et al., TaskExpert: Dynamically Assembling Multi-Task Representations with Memorial Mixture-of-Experts, ICCV, 2023b
>
> Elliot Meyerson and Risto Miikkulainen, Pseudo-task augmentation: From deep multitask learning to intratask sharing—and back, ICML, 2018.

---

### Official Review · Reviewer_boi3 · 2023-10-30

**Soundness:** 3 good
**Presentation:** 3 good
**Contribution:** 2 fair
**Rating:** 6
**Confidence:** 4

**Summary:**

The paper tackles the problem of multi-task learning for dense tasks like depth prediction, semantic segmentation, normal estimation. Cross-task correlations are noisy and there is no 3D regularization. The paper proposes introducing a latent that is structured and 3D-aware (by outputting a K-planes representation) and uses projection with differentiable rendering and a task specific decoder. The 3D awareness combined with view dependent rendering can be used to both regularize the 3D structure and multi-view consistency on the same 3D latent. Results are shown in NYUv2 and PASCAL-Context datasets.

**Strengths:**

The paper is quite well-written and easy to follow. The main idea of the paper is clear - to output a 3D feature representation from an image (using K-planes), which can then be used as part of a NeRF-like neural rendering pipeline to project the feature into a 2D feature image which can then be used by a task-specific decoder.

**Weaknesses:**

Predicting 3D representations (to use in neural rendering) from unlabeled 2D images and/or text is a relatively common idea [1, 2, 3, 4, 5]. The paper applies this idea in a multi-task setting, and adds a multi-view consistency for multi-view dataset. The results are not highly compelling compared to the baselines in the paper, which may internally learn some 3D structure too. So the explicit formulation of the 3D latent structure is not well motivated, unless the task requires some view editing or novel view synthesis, or an explicit 3D object representation, etc.

____
[1] Chan, Eric R., et al. "pi-gan: Periodic implicit generative adversarial networks for 3d-aware image synthesis." Proceedings of the IEEE/CVF conference on computer vision and pattern recognition. 2021.

[2] Cai, Shengqu, et al. "Pix2nerf: Unsupervised conditional p-gan for single image to neural radiance fields translation." Proceedings of the IEEE/CVF conference on computer vision and pattern recognition. 2022.

[3] Lin, Chen-Hsuan, et al. "Magic3d: High-resolution text-to-3d content creation." Proceedings of the IEEE/CVF Conference on Computer Vision and Pattern Recognition. 2023.

[4] Zhang, Jingbo, et al. "Text2NeRF: Text-Driven 3D Scene Generation with Neural Radiance Fields." arXiv preprint arXiv:2305.11588 (2023).

[5] Gu, Jiatao, et al. "Stylenerf: A style-based 3d-aware generator for high-resolution image synthesis." arXiv preprint arXiv:2110.08985 (2021).

**Questions:**

1. Its unclear if the results are statistically significant compared to the baseline. Since the method can be applied on any baseline MTL architecture, it is unclear if the improvements (especially in Table 2) are just due to randomness of the SGD training dynamics or is an actual improvement. Results over 3-5 trials should be reported if the improvements are marginal/inconsistent.

2. Table 3 shows that multiview consistency may not help in a MTL scenario considered in the paper. What is the significance of this section?

---

> ### Author Response · Authors · 2023-11-21
> **Response to Reviewer boi3**
>
> We thank the reviewer for the positive feedback and insightful comments. We address the questions and comments below.
>
> > Q1: ... 3D representations from ... 2D... is a relatively common idea [1, 2, 3, 4, 5]. ...The results are not highly compelling... not well motivated, unless the task requires some view editing...
>
> R1: Thanks for the suggested papers. We added them in the related work. Learning 3D representations has been shown to be crucial for view editing and novel view synthesis or explicit 3D object etc. as in the recommended papers [1,2,3,4,5]. This also motivates our work. Unlike these works, in this paper, we mainly focus on multi-task learning (MTL) which is one of the crucial aspects in computer vision and yet the majority of existing methods in MTL overlook the intrinsic 3D world. However, the physics behind 2D tasks is the 3D and the 3D space affords us inherent and implicit consistency across various 2D tasks, which is the key and helpful for learning a single multi-task network to jointly perform multiple dense prediction tasks. To this end, in our paper, we present an architecture-agnostic method that integrates a 3D-aware regularization to regulating the learned representation to be 3D-aware for tackling multiple vision dense prediction problems and demonstrate the benefits of leveragint the geometry for jointly tackling various 2D vision tasks without additional inference cost.
>
> In terms of the strength of our results, we show that our method can be incorporated to existing frameworks (CNN or ViT based models) and provides improvements for all tasks consistently, in comparison with the improvement from different state-of-the-art methods using the same backbone.  Also, it is challenging to achieve better performance in multi-task learning as it requires achieving better performance on all tasks instead of one task. In most multi-task dense prediction problems, the metric is not accuracy. For example, on NYUv2, the metric for segmentation is the mIoU computed over 40 categories and all pixels. Achieving +1.31 mIoU over the InvPT, a very competitive baseline, in NYUv2 without additional cost during inference, is a significant gain. To better demonstrate the comparisons between our method and the baselines in the context of multi-task learning, we report the multi-task learning performance as suggested in the prior (Vandenhende et al., 2021). Here the multi-task learning performance is computed as $\Delta\text{MTL} = \frac{1}{T} \sum_{t=1}^{T} (-1)^{\ell_t}(P_t - P_t^{baseline}) / P_t^{baseline}$, where $\ell_t=1$ if a lower value of $P_t$ means better performance for metric of task $t$, and 0 otherwise. Here, $\Delta\text{MTL}$ computes the gain w.r.t the baseline and is averaged by the number of task. As shown in the table below, in NYUv2, given the same backbone (HRNet48), the improvement achieved by our method is much more than the one achieved by the ATRC method (+2.17 vs +0.34). Also, we obtained +1.60 and +1.75 multi-task learning performance over the InvPT on NYUv2 and PASCAL benchmarks, respectively. This indicates that our method achieves relatively significant improvement over the baseline without additional cost in inference. Additionally, our framework provides a way to make most visual recognition method 3D aware and to be benefit from the geometry while having no additional inference cost.
>
> * Testing results on NYUv2
> |  Method | Backbone | Seg. (mIoU) | Depth (RMSE) | Normal (mErr) | Boundary (odsF) | $\Delta\text{MTL}$ |
> |:-------:|----------|:-----------:|:------------:|:-------------:|:---------------:|--------------------|
> | MTI-Net |  HRNet48 |    45.97    |    0.5365    |     20.27     |      77.86      |        0.00        |
> |   ATRC  |  HRNet48 |    46.33    |    0.5363    |     20.18     |      77.94      |        +0.34       |
> |   Ours  |  HRNet48 |  **46.67**  |  **0.5210**  |   **19.93**   |    **78.10**    |      **+2.17**     |
> |  InvPT  |   ViT-L  |    53.56    |    0.5183    |     19.04     |      78.10      |        0.00        |
> |   Ours  |   ViT-L  |  **54.87**  |  **0.5006**  |   **18.55**   |    **78.30**    |      **+1.60**     |
>
> * Testing results on PASCAL
> |  Method | Backbone | Seg. (mIoU) | PartSeg (mIoU) | Sal (maxF) | Normal (mErr) | Boundary (odsF) | $\Delta\text{MTL}$ |
> |:-------:|----------|:-----------:|:--------------:|:----------:|:-------------:|:---------------:|--------------------|
> | MTI-Net |  HRNet48 |    64.42    |      64.97     |    84.56   |     13.82     |      74.30      |        0.00        |
> |   Ours  |  HRNet48 |  **66.71**  |    **65.20**   |  **84.59** |   **13.71**   |    **74.50**    |      **+1.00**     |
> |  InvPT  |   ViT-L  |    79.03    |      67.61     |    84.81   |     14.15     |      73.00      |        0.00        |
> |   Ours  |   ViT-L  |  **79.53**  |    **69.12**   |  **84.94** |   **13.53**   |    **74.00**    |      **+1.75**     |

---

> > ### Author Response · Authors · 2023-11-21
> > **Response to Reviewer boi3**
> >
> > > Q2: Its unclear if the results are statistically significant compared to the baseline. Since the method can be applied on any baseline MTL architecture, it is unclear if the improvements (especially in Table 2) are just due to randomness of the SGD training dynamics or is an actual improvement. Results over 3-5 trials should be reported if the improvements are marginal/inconsistent.
> >
> > R2: Many thanks for the suggestion for reporting results over 3-5 trials. Results of state-of-the-art methods in our manuscript are directly from their original papers and they only report results of one run. We reported results of our method incorporated with InvPT over 3 runs on NYUv2 and PASCAL benchmarks below and included them in the supplementary. From the results shown below, we can see that our method is stable (i.e. the std is very small on each task) and improves over the baseline consistently on all tasks.
> >
> > Also, from the tables reported in the manuscript, we can see that the improvement of our method over the baseline is much greater than the improvement between two state-of-the-art methods using the same backbone (e.g. ATRC vs MTI-Net in Table 1). For example, in Table 1, the improvement of our method incorporated with MTI-Net over the MTI-Net is much greater than the improvement between ATRC and MTI-Net. In Table 2, our method obtains +2.29 mIoU on semantic segmentation over the MTI-Net, +1.51 mIoU on human parts segmentation and +1.00 odsF on boundary detection over InvPT and achieves better results than the baselines on all tasks consistently.
> >
> > * Testing results of our method over 3 runs on NYUv2
> > | Method |      Seg. (mIoU)     |       Depth (RMSE)      |     Normal (mErr)    |    Boundary (odsF)   |
> > |:------:|:--------------------:|:-----------------------:|:--------------------:|:--------------------:|
> > |  InvPT |         53.56        |          0.5183         |         19.04        |         78.10        |
> > |  Ours  | **54.86 $\pm$ 0.29** | **0.5000 $\pm$ 0.0010** | **18.49 $\pm$ 0.09** | **78.17 $\pm$ 0.09** |
> >
> > * Testing results of our method over 3 runs on PASCAL
> > | Method |      Seg. (mIoU)     |    PartSeg (mIoU)    |      Sal (maxF)      |     Normal (mErr)    |    Boundary (odsF)   |
> > |:------:|:--------------------:|:--------------------:|:--------------------:|:--------------------:|:--------------------:|
> > |  InvPT |         79.03        |         67.61        |         84.81        |         14.15        |         73.00        |
> > |  Ours  | **79.92 $\pm$ 0.32** | **69.08 $\pm$ 0.15** | **84.85 $\pm$ 0.06** | **13.70 $\pm$ 0.14** | **73.83 $\pm$ 0.17** |
> >
> > > Q3: Table 3 shows that multiview consistency may not help in a MTL scenario considered in the paper. What is the significance of this section?
> >
> > R3: In this work, we propose a 3D-aware regularizer to enable the multi-task learning network to learn geometric aware representations that improve the performance of multiple dense-prediction vision tasks. The regularizer does not rely on multiple views and cameras parameters and yet we provide a way of leveraging multi-views data, if they are available. From Table 3, the results suggest that better 3D geometry learning through multi-view consistency is beneficial, however, the improvements are modest. We argue that coarse 3D scene information obtained from single views can be sufficient to learn more structured and regulate inter-task relations and we believe Table 3 results are useful to convey this.

---

### Official Review · Reviewer_girT · 2023-10-31

**Soundness:** 3 good
**Presentation:** 2 fair
**Contribution:** 3 good
**Rating:** 6
**Confidence:** 4

**Summary:**

This paper proposed a structured 3D-aware regularizer that interfaces multiple tasks through the projection of features extracted from an image encoder to a shared 3D feature space and decodes them into their task output space through differentiable rendering. The experiment results are relatively good and promising.

**Strengths:**

Overall, I like this idea pretty much. For me, this work is like creating a new research line that makes most recognition methods 3D aware, which is dual to another research line that makes 2D image generation tasks 3D-aware. Although the introduced 3D-awared encoder does not explicitly give the 3D representation (I mean this method could not output decent 3D mesh), it gives way to investigating the 3D properties in most recognition tasks.

**Weaknesses:**

I paid a lot of expectations on the experiments after reading the Abstract and Introduction sections, but the experiments were not strong enough to match my expectations.

**Questions:**

For example,
1. how is the 3D quality of the learned nerf?
2. Any geometric insights about the performance improvement? otherwise, it will just be like a regularization paper (1 point improvement, that's all).
3. I think MTL might not be the best task to show the benefit of your method, do you have any ideas on that?

Overall, I think this is a decent paper to accept, and I know the questions I made above might not be easy to answer. I will increase my score if the authors can offer some convincing and motivated answers.

---

> ### Author Response · Authors · 2023-11-21
> **Response to Reviewer girT**
>
> We thank the reviewer for the positive feedback and insightful comments. We address the questions and comments below.
>
> > Q1: how is the 3D quality of the learned nerf?
>
> R1: We would like to emphasize that our goal is not to obtain very accurate 3D geometry of the scene but sufficiently accurate geometry to regulate the cross-task correlations by removing the ones implausible with 3D geometry  such as steep changes in a neighbourhood of a pixel while segmentation labels are uniform. In addition, we focused on achieving good training computation and performance tradeoff. In the paper, we show the results of outputs from the regularizer branch in Table 5 and they are also visualized in Figure 3. We can observe that the learned regularizer head can render reasonably high quality predictions. Also, as in Table 1, our method improves the performance on depth estimation, which coarsely evaluates the 3D quality. Additionaly, when we have multiple views of the same scene, the regularizer head is used to reconstruct the novel view based on the seen view, which is verified to be helpful in Table 3 and also evaluates the 3D quality.

---

> ### Author Response · Authors · 2023-11-21
> **Response to Reviewer girT**
>
> > Q2: Any geometric insights about the performance improvement? otherwise, it will just be like a regularization paper (1 point improvement, that's all).
>
> R2: Multi-task learning can be considered one of the crucial aspects in computer vision and yet the majority of existing methods overlook the intrinsic 3D world, which affords us the inherent and implicit consistency between various computer vision tasks. To this end, we proposed a framework that adds the 3D-aware regularization term to regularize the shared representations to be 3D-aware for achieving better cross-task consistency and more accurate predictions. In the regularization, outputs for all tasks are conditioned on observations that lie on a low-D manifold (the density (Mildenhall et al., 2020)), enforcing 3D consistency between tasks. In Figure 3, we visualize the predictions of four tasks in NYUv2 and our method can be observed to estimate better predictions consistently for four tasks. For example, our method estimates more accurate predictions around the boundary of the refrigerator, stove and less noisy predictions within objects like the curtain and stove. The geometric information learned in our method helps distinguish different adjacent objects, avoids noisy predictions at object boundaries and also improves the consistency across tasks as all tasks predictions are rendered based on the same density in the regularizer.
>
> In terms of the strength of our results, we show that our method can be incorporated to existing frameworks (CNN or ViT based models) and provides improvements for all tasks consistently, in comparison with the improvement from different state-of-the-art methods using the same backbone.  Also, it is challenging to achieve better performance in multi-task learning as it requires achieving better performance on all tasks instead of one task. In most multi-task dense prediction problems, the metric is not accuracy. For example, on NYUv2, the metric for segmentation is the mIoU computed over 40 categories and all pixels. Achieving +1.31 mIoU over the InvPT, a very competitive baseline, in NYUv2 without additional cost during inference, is a significant gain. To better demonstrate the comparisons between our method and the baselines in the context of multi-task learning, we report the multi-task learning performance as suggested in the prior (Vandenhende et al., 2021). Here the multi-task learning performance is computed as $\Delta\text{MTL} = \frac{1}{T} \sum_{t=1}^{T} (-1)^{\ell_t}(P_t - P_t^{baseline}) / P_t^{baseline}$, where $\ell_t=1$ if a lower value of $P_t$ means better performance for metric of task $t$, and 0 otherwise. Here, $\Delta\text{MTL}$ computes the gain w.r.t the baseline and is averaged by the number of task. As shown in the table below, in NYUv2, given the same backbone (HRNet48), the improvement achieved by our method is much more than the one achieved by the ATRC method (+2.17 vs +0.34). Also, we obtained +1.60 and +1.75 multi-task learning performance over the InvPT on NYUv2 and PASCAL benchmarks, respectively. This indicates that our method achieves relatively significant improvement over the baseline with no additional cost during inference.
>
>
> * Testing results on NYUv2.
> |  Method | Backbone | Seg. (mIoU) | Depth (RMSE) | Normal (mErr) | Boundary (odsF) | $\Delta\text{MTL}$ |
> |:-------:|----------|:-----------:|:------------:|:-------------:|:---------------:|--------------------|
> | MTI-Net |  HRNet48 |    45.97    |    0.5365    |     20.27     |      77.86      |        0.00        |
> |   ATRC  |  HRNet48 |    46.33    |    0.5363    |     20.18     |      77.94      |        +0.34       |
> |   Ours  |  HRNet48 |  **46.67**  |  **0.5210**  |   **19.93**   |    **78.10**    |      **+2.17**     |
> |  InvPT  |   ViT-L  |    53.56    |    0.5183    |     19.04     |      78.10      |        0.00        |
> |   Ours  |   ViT-L  |  **54.87**  |  **0.5006**  |   **18.55**   |    **78.30**    |      **+1.60**     |
>
>
> * Testing results on PASCAL.
> |  Method | Backbone | Seg. (mIoU) | PartSeg (mIoU) | Sal (maxF) | Normal (mErr) | Boundary (odsF) | $\Delta\text{MTL}$ |
> |:-------:|----------|:-----------:|:--------------:|:----------:|:-------------:|:---------------:|--------------------|
> | MTI-Net |  HRNet48 |    64.42    |      64.97     |    84.56   |     13.82     |      74.30      |        0.00        |
> |   Ours  |  HRNet48 |  **66.71**  |    **65.20**   |  **84.59** |   **13.71**   |    **74.50**    |      **+1.00**     |
> |  InvPT  |   ViT-L  |    79.03    |      67.61     |    84.81   |     14.15     |      73.00      |        0.00        |
> |   Ours  |   ViT-L  |  **79.53**  |    **69.12**   |  **84.94** |   **13.53**   |    **74.00**    |      **+1.75**     |
>
>
> Vandenhende et al., Multi-task learning for dense prediction tasks: A survey, PAMI, 2021.

---

> > ### Author Response · Authors · 2023-11-21
> > **Response to Reviewer girT**
> >
> > > Q3: I think MTL might not be the best task to show the benefit of your method, do you have any ideas on that?
> >
> > R3: In this paper, we focus on jointly performing multiple dense prediction tasks and improving the performance by regulating the shared representations to be 3D-aware. In our method, multiple tasks signals can help learn better geometry, like depth estimation, while in turn the geometry can help multiple dense prediction tasks, like segmentation in boundary (better separating objects as shown in Figure 3). As mentioned by the reviewer, our framework is applicable to most vision tasks including as object detection, object tracking, corresponding points by making them 3D aware and benefit these tasks with no additional inference cost.

---

### Official Review · Reviewer_aMPL · 2023-11-01

**Soundness:** 3 good
**Presentation:** 3 good
**Contribution:** 3 good
**Rating:** 6
**Confidence:** 3

**Summary:**

This paper proposes to improve multi-task dense prediction by introducing a 3D representation branch during training. This 3D-aware branch(a triplane representation with volumetric render) shares the same backbone as a conventional multi-task learning setup. This NeRF branch encourages the feature extracted from the backbone to contain rich 3D information and to follow the physical imaging process.
 This 3D-aware head also enables supervising with multi-view consistency. Experiments show that this auxiliary 3D aware branch helps improve the performance of the conventional branch during inference when the 3D aware branch is dropped.

**Strengths:**

(1) Interesting idea, the auxilary 3D aware branch  forces the backbone to be truly 3D-aware. This method also improves the alignment between different prediction heads and the depth/density head as they will be rendered following the physical imaging process.

(2) Using 3D aware representation as an auxiliary branch only for regularization also enables fast inference, which is a new and interesting idea to me.

**Weaknesses:**

(1) The 3D aware branch uses Triplane representation and volumetric render, which needs a specific camera model (intrinsic matrix). So it is suspicious that the model can somehow overfit to the specific camera parameters.  As a comparison, pixel-aligned scene representation (i.e., PiFU) can resolve this problem and it uses NDC representation.
For scenes with different camera parameters, the performance is unclear.

(2) Although the tri-plane representation has significantly reduced the memory cost and rendering of NeRF. Rendering the whole image and calculating the gradient is still a heavy burden. Common practice includes random sampling or pixel binning.  In the supplement material, there is an incomplete explanation. It is important to clarify the NeRF sampling implementation and extra training cost(memory footprint and FLOPS)

**Questions:**

My main concerns are (1) the potential camera overfitting problem, and (2) The extra training cost introduced by the NeRF head.

---

> ### Author Response · Authors · 2023-11-21
> **Response to Reviewer aMPL**
>
> We thank the reviewer for the positive feedback and insightful comments. We address the questions and comments below.
>
> > Q1: ...the potential camera overfitting problem...
>
> R1: Thanks for the recommended papers and the comment.  In our paper, the 3D coordinates and strategy of projecting the 3D coordinates onto the feature planes are similar to the ones in PiFU (Saito et al., 2019) and NDC (Yao et al., 2023). The feature planes are generated by the feature encoder and it is pixel-wise feature map instead of a global pooled feature vector. The $x$ and $y$ coordinates are aligned with pixel locations and $z$ is the depth value. We follow Chan et al. (2022) that projects the coordinates $(x, y, z)$ onto three planes $e_{xy}, e_{yz}, e_{xz}$, retrieving the features via bilinear interpolation, and aggregates features from three planes instead of taking the 2D features and the $z$ values as representations in PiFU (Saito et al., 2019) or dividing the dimension of the feature map channel into $D$ groups ($D$ is the number of depth bins) in  NDC (Yao et al., 2023). So our method has similar property as in PiFU (Saito et al., 2019) and NDC (Yao et al., 2023) and does not overfit to the camera parameters. We include this discussion in the supplementary.
>
> Furthermore, as we first feed the image into the feature encoder, which scales the 3D coordinates accordingly and the coordinates will not be absolute but at the right scale for rendering. After training, the 3D-aware regularizer is discarded, the remaining the multi-task learning branch does not require camera parameters for generating predictions for different tasks. In addition, we visualize multiple images' predictions of the regularizers on PASCAL in Figure 1 in the supplementary. The PASCAL dataset consists of annotated consumer photographs collected from the flickr photo-sharing web-site, taken by various cameras with different intrinsics. From Figure 1 shown in the supplementary, we can see that the regularizer can render good quality predictions for all tasks on all images which also indicates that it does not overfit to the camera parameters.
>
>
> Saito et al., PIFu: Pixel-Aligned Implicit Function for High-Resolution Clothed Human Digitization, ICCV, 2019.
>
> Yao et al., NDC-Scene: Boost Monocular 3D Semantic Scene Completion in Normalized Device Coordinates Space, ICCV, 2023.
>
> > Q2: ... NeRF sampling implementation and extra training cost(memory footprint and FLOPS)
>
> R2: Thanks for the suggested techniques for reducing the cost, including random sampling, pixel binning. We included more details about the implementation and discussed the recommended techniques in the future work section in the supplementary. We will make our code public for the final version.
>
> For implementing the tri-plane representation, following Chan et al. (2022), we map the shared representations to three feature planes and we render small resolution predictions for all tasks. We further use a two-pass importance sampling as in NeRF (Mildenhall et al., 2020). For the majority of the experiments in the manuscript, we use 128 total depth samples per ray. We then project each sampled 3D coordinates onto the tri-plane, retrieving the features and aggregating the features from the tri-plane. Finally, we render 56x72 predictions for NYUv2 and 64×64 for PASCAL-Context and resize the predictions via bilinear interpolation to the groundtruth resolution.
>
> We included the comparisons of memory and computational cost during training in the supplementary. Our method that incorporates the regularizer to the MTL baseline slightly increases the number of parameters (Ours vs InvPT: 1.016 vs 1) and FLOPS (Ours vs InvPT: 1.114 vs 1) during training, training time (Ours vs InvPT: 1.318 vs 1), and training memory (Ours vs InvPT: 1.397 vs 1). We highlight that there is *NO additional cost* during inference, since the regularizer will be discarded during inference.
>
> Chan et al., Efficient geometry-aware 3d generative adversarial networks, CVPR, 2022.
>
> Mildenhall et al., NeRF: Representing scenes as neural radiance fields for view synthesis, ECCV, 2020

---

> ### Comment · Reviewer_aMPL · 2023-12-05
> **Post rebuttal comment**
>
> I've read the response and also comments from other reviewers. The idea of regularizing detection feature maps with a rendering branch is novel, interesting, and inspiring.
>
> Q1, the camera parameter overfitting problem. The authors have partially addressed my concern (but not fully), as the NeRF/rendering branch is discarded during the reference, which brings little problem for the inference stage. However, any direct 3D lifting method such as rearranging the ConvNet features to tri-plane encodes the camera parameters implicitly in the feature extraction stage. Notice that only NDC is a representation that is camera parameter-agnostic.
>
> Q2 has also been clarified.
>
> I still keep my original rating,i.e.,  6: marginally above the acceptance threshold

---

### Author Response · Authors · 2023-11-21
**Response to all Reviewers**

We thank the reviewers for their valuable time and feedback, and for acknowledging the clear writing (Reviewer boi3, 4RSv), interesting idea (Reviewer aMPL, girT), significant contributions (Reviewer girT, 4RSv). We address the questions and comments below and we will make our code publicly available for the final version. We have uploaded a revised version of the manuscript (highlighted in blue).

---

### Meta-Review · Area_Chair_9PfU · 2023-12-10

**Metareview:**

(a) The paper presents a novel approach to multi-task dense prediction, introducing a 3D-aware branch during training. This branch, employing a triplane representation and volumetric rendering, shares the same backbone with the existing multi-task learning model. The primary claim is that this 3D-aware branch enriches feature extraction with 3D information and adheres to the physical imaging process, thus improving multi-task learning performance. This approach also enables supervision with multi-view consistency.

(b) The paper's primary strength lies in its novel approach, particularly the introduction of an auxiliary 3D-aware branch for training. This represents a substantial deviation from existing methods, demonstrating moderate performance improvements compared to the baseline. Another strength is that the 3D-aware representation is solely for regularization, allowing for fast inference without additional computational costs during this stage.

(c) Weaknesses: Despite its innovative nature, the application of the 3D-aware regularizer for single-view supervision is not fully justified. The paper does not demonstrate that the regularizer learns meaningful 3D representations. Merely presenting the rendered RGB images is insufficient; a more thorough examination of the rendered depth, normal maps, and rendered RGB from novel views would be more convincing. Additionally, in the updated supplementary material, the paper shows a performance gap when compared to more recent methods, which is either marginal or indicates inferior performance. This raises questions about the method's contribution to the field.

**Justification For Why Not Higher Score:**

Despite its innovative nature, the application of the 3D-aware regularizer for single-view supervision is not fully justified. The paper does not demonstrate that the regularizer learns meaningful 3D representations. Merely presenting the rendered RGB images is insufficient; a more thorough examination of the rendered depth, normal maps, and rendered RGB from novel views would be more convincing. Additionally, in the updated supplementary material, the paper shows a performance gap when compared to more recent methods, which is either marginal or indicates inferior performance. This raises questions about the method's contribution to the field.

**Justification For Why Not Lower Score:**

All reviewers give borderline accept ratings. The paper's primary strength lies in its novel approach, particularly the introduction of an auxiliary 3D-aware branch for training. This represents a substantial deviation from existing methods, demonstrating moderate performance improvements compared to the baseline. Another strength is that the 3D-aware representation is solely for regularization, allowing for fast inference without additional computational costs during this stage.

---

### Decision · Program_Chairs · 2024-01-16

Accept (poster)